ZebraShare: a new venue for rapid dissemination of zebrafish mutant data

DeLaurier April AprilD@usca.edu 1
Howe Douglas G. 2
Ruzicka Leyla 2
Carte Adam N. 3 4 5
Mishoe Hernandez Lacie 1
Wiggins Kali J 1
Gallati Mika M. 6
Vanpelt Kayce 1
Loyo Rosado Frances 1
Pugh Katlin G. 1
Shabdue Chasey J. 1
Jihad Khadijah 1
Thyme Summer B. 7
Talbot Jared C. jared.talbot@maine.edu 6
1 Department of Biology and Geology, University of South Carolina –Aiken , Aiken , SC , United States of America
2 The Institute of Neuroscience, University of Oregon , Eugene , OR , United States of America
3 Department of Molecular and Cellular Biology, Harvard University , Cambridge , MA , United States of America
4 Systems, Synthetic, and Quantitative Biology Program, Harvard University , Cambridge , MA , United States of America
5 Biozentrum, Universität Basel , Basel , Switzerland
6 School of Biology and Ecology, University of Maine , Orono , ME , United States of America
7 Department of Neurobiology, University of Alabama –Birmingham , Birmingham , AL , United States of America
Posner Mason
Electronic publication date: 2021 Apr 13
Publication date: 2021
Volume: 9
Electronic Location ID: e11007
Received 2020 Sep 3; Accepted 2021 Feb 2
Copyright: ©2021 DeLaurier et al.
Copyright year: 2021
Copyright holder: DeLaurier et al.
License: This is an open access article distributed under the terms of the Creative Commons Attribution License, which permits unrestricted use, distribution, reproduction and adaptation in any medium and for any purpose provided that it is properly attributed. For attribution, the original author(s), title, publication source (PeerJ) and either DOI or URL of the article must be cited.
License URL: https://creativecommons.org/licenses/by/4.0/

Keywords: Zebrafish, nhp2l1, lsd1, kdm1a, snu13, phf21a, ctnnd1, Collaboration

Funding: University of Maine new investigator startup funds NIH T32 NS077984 NIH GM088041 GM117964 University of South Carolina Aiken startup funds University of South Carolina RISE and ASPIRE-I awards PROBe awards and a Developmental Research Program grant through NIH/NIGMS SC INBRE P20GM103499 NIH MH110603 National Human Genome Research Institute at the National Institutes of Health U41 HG002659 (ZFIN) U24 HG010859 This research is supported by University of Maine new investigator startup funds, an NIH T32 training grant NS077984 (to Jared C. Talbot), and by NIH grants GM088041 and GM117964 (to Sharon L Amacher). This work was also supported by University of South Carolina Aiken startup funds, University of South Carolina RISE and ASPIRE-I awards, and undergraduate PROBe awards and a Developmental Research Program grant through NIH/NIGMS SC INBRE P20GM103499 (to April DeLaurier). This work was also supported by NIH grant MH110603 (to Summer B. Thyme). ZFIN is supported by the National Human Genome Research Institute at the National Institutes of Health [U41 HG002659 (ZFIN) and U24 HG010859 (Alliance of Genome Resources)]. The funders had no role in study design, data collection and analysis, decision to publish, or preparation of the manuscript.

==============================
Background

In the past decade, the zebrafish community has widely embraced targeted mutagenesis technologies, resulting in an abundance of mutant lines. While many lines have proven to be useful for investigating gene function, many have also shown no apparent phenotype, or phenotypes not of interest to the originating lab. In order for labs to document and share information about these lines, we have created ZebraShare as a new resource offered within ZFIN.

Methods

ZebraShare involves a form-based submission process generated by ZFIN. The ZebraShare interface (https://zfin.org/action/zebrashare) can be accessed on ZFIN under “Submit Data”. Users download the Submission Workbook and complete the required fields, then submit the completed workbook with associated images and captions, generating a new ZFIN publication record. ZFIN curators add the submitted phenotype and mutant information to the ZFIN database, provide mapping information about mutations, and cross reference this information across the appropriate ZFIN databases. We present here examples of ZebraShare submissions, including phf21aa, kdm1a, ctnnd1, snu13a, and snu13b mutant lines.

Results

Users can find ZebraShare submissions by searching ZFIN for specific alleles or line designations, just as for alleles submitted through the normal process. We present several potential examples of submission types to ZebraShare including a phenotypic mutants, mildly phenotypic, and early lethal mutants. Mutants for kdm1a show no apparent skeletal phenotype, and phf21aa mutants show only a mild skeletal phenotype, yet these genes have specific human disease relevance and therefore may be useful for further studies. The p120-catenin encoding gene, ctnnd1, was knocked out to investigate a potential role in brain development or function. The homozygous ctnnd1 mutant disintegrates during early somitogenesis and the heterozygote has localized defects, revealing vital roles in early development. Two snu13 genes were knocked out to investigate a role in muscle formation. The snu13a;snu13b double mutant has an early embryonic lethal phenotype, potentially related to a proposed role in the core splicing complex. In each example, the mutants submitted to ZebraShare display phenotypes that are not ideally suited to their originating lab’s project directions but may be of great relevance to other researchers.

Conclusion

ZebraShare provides an opportunity for researchers to directly share information about mutant lines within ZFIN, which is widely used by the community as a central database of information about zebrafish lines. Submissions of alleles with a phenotypic or unexpected phenotypes is encouraged to promote collaborations, disseminate lines, reduce redundancy of effort and to promote efficient use of time and resources. We anticipate that as submissions to ZebraShare increase, they will help build an ultimately more complete picture of zebrafish genetics and development.

Introduction

In the last decade, use of reverse genetics has become a standard approach to investigate gene function in zebrafish and other species. With the advent of zinc finger nucleases it became possible to direct mutagenesis in zebrafish, which became easier with TALENs and then even simpler with CRISPR-Cas9 technology (Rafferty & Quinn, 2018). The simplicity of targeted mutagenesis in zebrafish has led to mass production of knockout lines targeting genes and genetic pathways of interest. Some mutants show phenotypes that have led to impactful publications, but it is apparent that many more mutants have no phenotypic defect (a phenotypic), subtle phenotypes, or phenotypes in tissues that are not in the research focus of the originating lab (Kok et al., 2015; Stainier, Kontarakis & Rossi, 2015). Although mutants with unexpected phenotypes may sometimes not be pursued by the originating lab, information about these lines is still critically relevant to the broader research community. Failure to disseminate these findings and alleles will ultimately lead to redundant efforts, lost time, and wasted resources. To facilitate information distribution about zebrafish lines, we have integrated a new feature into the Zebrafish Information Network (ZFIN) called “ZebraShare”. In ZebraShare, users submit an abstract, knockout sequences, validation steps, and phenotypic information directly to ZFIN. Submissions are curated by ZFIN staff into the ZebraShare database for public viewing. Future information or edits can be added to the submission over time, which may include further descriptions of phenotypes or other relevant information about lines. We anticipate that ZebraShare will help zebrafish researchers engage in optimized use of their reverse genetics mutants by avoiding redundancy, sharing phenotypes that would be otherwise lost, and forging collaborations for future research. Here, we describe the ZebraShare feature of ZFIN. We begin by describing the submission process and features included which ensure that data quality can quickly be assessed by ZFIN users. Then, we provide four differing examples of submissions and for each example provide the rationale for constructing the mutant and why the resulting phenotypes lead to ZebraShare submission. Finally, we discuss the implications of this new sharing system.

Materials & Methods

Animal stocks and husbandry

We raised and housed zebrafish in standard conditions (Westerfield, 2007) and collected embryos by natural spawning of adult fish, with embryo staging as described (Kimmel et al., 1995). All zebrafish experimentation was conducted as approved by Institutional Animal Care and Use Committees at the University of Maine (approval number A2019_10_01), The Ohio State University (approval number 2012A00000113), the University of Alabama at Birmingham (approval number 21744), Harvard University (approval number 25-08), the University of South Carolina (approval number 2485-101478-031720) and the University of South Carolina Aiken (010317-BIO-01).

Oligonucleotides

Table 1 lists the oligonucleotides used in this study during mutant construction and genotyping.

Table 1 gRNA targets used for CRISPR and genotyping primer sequences (5′–3′).

These mutants were generated in three separate labs so they used different mutagenesis and genotyping protocols. Three gRNAs were co-injected for ctnnd1.

Originating lab	Gene	gRNA target(s)	Forward Genotyping Primer	Reverse genotyping primer	
DeLaurier	phf21aa	GTGAGGCTAGCAGCAGGCAG	T7 assay: GATTTCCTTGCCACTAGCAC	T7 assay: CCATTAAGAAGCAGCACAGG	
Traditional genotyping: AGAATACTGTTGGCCTCCTG	Traditional genotyping: CCATTAAGAAGCAGCACAGG	
DeLaurier	kdm1a	GGCTCCTCCTCTTCGTCAGG	T7 assay/Traditional genotyping: AAGGAGAAGCCTCTGTCATC	T7 assay/Traditional genotyping: GAGATGTTTACCTTTGCCCG	
Thyme	ctnnd1	GGTCCAACTGAGGTCGGCTG, CCTCCAGGCCATAGGGCTCT, CTGATCGTCCTCCAGGCCAT	Traditional genotyping: ATGGCTACCGCACGCTGGAC	Traditional genotyping: GTGTGGATGTGCCAACCGGGG	
Talbot	snu13a	GAACCCTAAAGCGTACCCTC	HRMA genotyping: GACTGATCAAGTGCTGTTCTCC	HRMA genotyping: ATCCAGGATGGTTTTGCTGAGG	
DNA sequencing: TGGCTAATCTTTATGGTTCAGG	DNA sequencing: CTTCGTTGGCCCCTTTC	
Talbot	snu13b	GAACCCTAAAGCCTATCCTC	HRMA genotyping: GTCTGTGGTTTTTACTCAGACTG	HRMA genotyping: CCCCTTTTCTCAGCTGTTTG	
DNA sequencing: TGCTAACCGGATGATAAGAG	DNA sequencing: CGAGTTATTCACCTTCATTGG	

phf21aa mutant construction

For phf21aa mutant (phf21aaaik4) construction, wild-type AB embryos were co-injected with 3 nl of a mixture containing guide RNA targeting exon 6 (ENSDART00000173629.2) (∼160 ng/µl) along with mRNA encoding nuclear-localized Cas9 (∼160 ng/µl). Nuclear-localized Cas9 mRNA was synthesized from pCS2-nCas9n (Addgene), linearized with NotI-HF (New England Biolabs), column purified (Zyppy Plasmid Miniprep kit; Zymo Research), and mRNA was synthesized (mMessenge mMachine SP6 kit; Thermo Fisher). Mutagenesis efficiency was detected in groups of F0 embryos (5 pooled × 3 replicates) using T7 endonuclease (New England Biolabs) digest of PCR fragments flanking the gRNA target site (PCR product = 995 base-pairs (bp), digestion products are approximately 720 and 275 bp). DNA from potential individual mutants was Sanger sequenced to establish a line with a 7 bp deletion at the gRNA target site in exon 6 ( phf21aaaik4). This mutation results in a frameshift mutation producing extensive missense and a premature stop codon (GenBank accession numbers: wild type MW438986, mutant MW438985). Genotyping of subsequent individual phf21aaaik4 fish utilized primers flanking the InDel site. PCR amplification results in a 641 bp product for wild-type DNA and a 634 bp product for mutant DNA. PCR products are run on a 2.5% agarose gel to resolve bands (wild type = 641 bp, mutant = 634 bp, heterozygotes = 641 + 634 bp bands).

kdm1a mutant construction

For kdm1a mutant (kdm1aaik5) construction, wild-type AB embryos were co-injected with guide RNA targeting exon 1 (ENSDART00000180532.1) (∼200 ng/µl) along with mRNA encoding nuclear-localized Cas9 (∼160 ng/µl). Nuclear-localized Cas9 mRNA was synthesized and injected as described above for phf21aa. Mutagenesis efficiency was detected in groups of F0 embryos (5 pooled × 3 replicates) using T7 endonuclease digest of PCR fragments flanking the guide RNA target site (PCR product = 299 bp, digestion products are approximately 228 and 71 bp), as described for phf21aa. cDNA from individual potential mutants was Sanger sequenced to establish a line with a 14 bp deletion at the guide RNA target site in exon 1 (kdm1aaik5). This mutation is predicted to result in a frameshift producing extensive missense and a premature stop codon. Genotyping of subsequent individual kdm1aaik5 fish utilizes the same T7 primers flanking the InDel site. PCR amplification results in a 299 bp product for wild-type DNA and a 285 bp product for mutant DNA. PCR products are run on a 2.5% agarose gel to resolve bands (wild type = 299 bp, mutant = 285 bp, heterozygotes = 299 + 285 bp bands).

ctnnd1 mutant construction

The ctnnd1 mutant (ctnnd1uab302) was constructed by injection of three guide RNAs (>50 ng/µl each) and purified Cas9 protein (25 µM) into wild-type EKW embryos. The first two nucleotides of every guide were changed to 5′-GG- 3′ for high-yield synthesis with T7 polymerase. Heterozygous carriers were initially identified on pools with MiSeq sequencing and later confirmed with Sanger sequencing. PCR of this 31 bp deletion, results in a 242 bp product for wild-type DNA and a 211 bp product for mutant DNA. PCR products were separated with standard agarose gel electrophoresis on 4% gels. The injected (F0) fish were raised to adulthood and F1 carriers confirmed by sequencing were outcrossed to wild-type EKW fish.

snu13a and snu13b mutant construction

The snu13aoz24 and snu13boz91 mutants were constructed following described methods (Talbot & Amacher, 2014). An injection mix containing 38 ng/µl of guide RNA targeting snu13a and 83 ng/µl of mRNA encoding nuclear localized Cas9 (Jao, Wente & Chen, 2013) was injected into AB fish. For snu13b, the injection was similar except the snu13b guide RNA had a concentration of 37 ng/µl. To prepare the Cas9 mRNA we synthesized from pCS2-nCas9n (Jao, Wente & Chen, 2013) after linearization with NotI (New England Biolabs); we used a mMessage mMachine kit (Thermo Fisher) to transcribe mRNA and purified the transcripts using a NucleoSpin II RNA cleanup kit (Machery-Nagel). Mutagenesis efficiency was determined using High Resolution Melt Analysis (HRMA) (Talbot & Amacher, 2014; Dahlem et al., 2012). These F1 carriers were outcrossed and identified by testing 16 embryos per clutch using HRMA. HRMA was again used to pre-screen F1 heterozygote carriers, which were sequenced using primers specific to snu13a and snu13b. Sequence analysis was performed on individual snu13a−∕−; snu13b−∕− embryos.

Histological staining and imaging of skeletal tissue

Larval skeletal samples (phf21aa and kdm1a) were prepared and stained using Alcian Blue and Alizarin Red dyes as described (Walker & Kimmel, 2007; DeLaurier, Alvarez & Wiggins, 2019). Samples were flat-mounted and imaged using an Olympus BX41 compound microscope and Olympus cellSens Standard software (version 1.16).

DAPI-stained embryos

ctnnd1 embryos were left in the chorion and fixed overnight in 4% formaldehyde in PBS. Embryos were then washed 4 × 5 min in PBS, incubated in DAPI for 30 min, and washed 2 × 5 min in PBS before being mounted in a droplet of 1% low-melting agarose in PBS on a 35 mm MatTek dish with a No. 1.5 coverslip bottom. Imaging was performed on a Zeiss LSM700 inverted laser scanning confocal microscope with a Plan-Apochromat 10X/0.45 air objective using 5 µm slices. Maximum intensity projections were produced from acquired z-stacks in Fiji (Schindelin et al., 2012), and images were scaled to maximize for visibility.

Figure 1 Publicizing a mutant on ZebraShare in 5 steps.

Figure 2 Example of how to determine DNA coordinates for a simple deletion allele, phf21aaaik4.

(A) Align the WT and mutant sequence. (B) Blast the aligned WT sequence and determine the base numbers altered in the mutant. (C) Transfer these coordinates to the ZebraShare submission workbook. Descriptions become more complex for combined insertion/deletions alleles (InDels) and for alleles with multiple mutation sites due to use of multiple guide RNAs.

Live imaging snu13 mutants

Zebrafish embryos from in-cross of snu13aoz24∕+; snu13boz91∕+ were monitored through their first 12 h of development, and then imaged using a Leica DMC5400 camera mounted on a Leica MZ10F microscope at 24 h post-fertilization (hpf).

Live imaging ctnnd1 mutants

ctnnd1 embryos were photographed at approximately the 6-somite stage using identical magnification and lighting settings across embryos on a Zeiss AXIO Zoom V16 microscope fitted with a PlanNeoFluar Z 1x/0.25 objective and Axiocam 503 color camera. Embryo photographs were color-balanced using the BIOP SimpleColorBalance plugin in ImageJ (Schindelin et al., 2012). The time-lapse recording of developing ctnnd1 embryos was made from approximately the 4–6 somite stage to the 12–14 somite stage using the “TIME-LAPSE” function on an iPhone 8 mounted to a Zeiss Stemi 2000 stereo microscope with a Gosky Universal Cell Phone Adapter Mount.

Results

ZebraShare implementation

ZebraShare implements several new features into existing ZFIN functionalities. A ZebraShare landing page, linked from the “Submit Data” button on the ZFIN home page, contains a summary of the project and links to the workbook and submission page (Fig. 1). ZebraShare is designed to fit into ZFIN’s existing publication acquisition infrastructure, nomenclature, and curation workflows. When completing the ZebraShare Workbook, researchers are asked to define precise coordinates of the mutation so the alleles can be described accurately in ZFIN (Fig. 2) and to provide information relevant to the mutant phenotypes. When uploading this workbook, authors have an opportunity to enter an abstract describing their allelic and phenotypic characterization (Fig. 3, Table 2). This abstract is linked to the workbook and any images and captions included in the submission. Upon creation, ZebraShare submissions are automatically assigned to the ZFIN nomenclature coordinator. The coordinator vets nomenclature, consults authors if needed, and after adding alleles to ZFIN with correct nomenclature, assigns the paper to the high-priority ZebraShare curation queue. Curators complete curation by adding the remaining details for mutants to the publication and inform the authors that their submission has been curated. Once a ZebraShare submission is completed, the mutant alleles and phenotypes can be searched for just like the ZFIN entries curated from papers (Van Slyke et al., 2018).

Figure 3 Example of a mutant, phf21aaaik4, publicized via ZebraShare.

(A) The allele description page shows information that helps researchers interpret the mutant. (B) The abstract page gives an overview of phenotypic characterization. (C) The figure associated with the abstract shows pertinent phenotypic details. For phf21aaaik4, the mutants appeared normal, save for a mild rotation of the ceratohyal cartilage (red arrow). Fish larvae are stained with alcian blue (cartilage) and alizarin red (bone) to reveal skeletal shape. The three scalebars in C each = 200 microns.

Ensuring validated mutant information

ZebraShare provides researchers an opportunity to detail their own validation steps in the submission workbook, which will be listed on the allele page. First, researchers enter detailed descriptions of the mutation using a written description, sequence alignment, and predicted effect on transcript and protein. Then, researchers specify whether the transcript changes are determined directly from cDNA sequencing or inferred from genomic sequences. A field is also provided where researchers can specify whether nonsense-mediated decay (NMD) has been assayed, because of the growing concern that compensation may occur in alleles that induce NMD (Rossi et al., 2015; El-Brolosy et al., 2019). Researchers can state whether mutations have been examined in homozygous embryos from heterozygous parents, or whether maternal-zygotic knockouts have been examined. Information about maternal-zygotic outcomes may be particularly important for mutations showing little or no phenotype in the offspring of heterozygous crosses. While validation experiments are not required for submission to ZebraShare, these fields are included to provide researchers an opportunity to communicate this information if desired. Because mutant validation may be improved after submission, and line availability may change, the following fields remain editable after submission: ‘Functional Consequence’, ‘Adult Viable’, ‘Maternal Zygosity Examined’, ‘NMD Apparent’, ‘Other Line Information’, and ‘Available’. Only the submitting researcher and other researchers designated at the time of mutant allele submission are able to edit these fields.

Table 2 Web addresses for the ZebraShare abstracts for alleles used in case studies.

Allele(s)	ZebraShare abstract page	
phf21aaaik4	http://zfin.org/ZDB-PUB-190605-16	
kdm1aaik5	http://zfin.org/ZDB-PUB-200515-15	
ctnnd1uab302	http://zfin.org/ZDB-PUB-200621-10	
snu13aoz24, snu13boz91	http://zfin.org/ZDB-PUB-200604-17	

Example 1, phf21aa knockout shows a mild craniofacial skeletal phenotype

phf21aa homozygous mutants develop normally, have no obvious external abnormalities, and are viable, fertile adults. A slight medial rotation of the ceratohyal cartilage was detected in whole mount mutant specimens at 7 days post fertilization (dpf) (Fig. 3C) in maternal-zygotic mutants (7/20) but not detected in heterozygote offspring (35/35). Flat mount of pharyngeal skeletons reveals no ceratohyal patterning defect in mutants compared to wild-type siblings (Fig. 3C), suggesting the rotation defect may be the result of a connective tissue defect not apparent in skeletal preparations. Loss of PHF21A is associated with Potocki-Shaffer Syndrome (PSS) in humans and is associated with craniofacial and neurological complications (Kim et al., 2012; Kim et al., 2019). Thus, although this mutant did not have a skeletal phenotype of interest to the originating lab, the phf21aa line may have other important uses as a disease model, so information about this line was provided to ZebraShare (Mishoe & DeLaurier, 2020). Researchers with interest in pursuing a zebrafish model for PSS may wish to investigate the origin of the anatomical defect in phf21aa mutants, in double mutants for the zebrafish co-ortholog for phf21aa, phf21ab, or in combination with mutants for other interacting factors (i.e., kdm1a, ZNF198/zmym2, ZNF261/zmym3) (Hakimi et al., 2003; Shi et al., 2004; Lan et al., 2007; Kim et al., 2012; Kim et al., 2019).

Example 2, kdm1a mutants have no overt skeletal phenotype

kdm1a (zygotic and maternal-zygotic) homozygous mutants develop normally, have no obvious external abnormalities, and are viable, fertile adults. Analysis of craniofacial skeletal patterning in kdm1a maternal-zygotic mutants at stages between 4–8 dpf (Fig. 4) reveals no specific defects in cartilage or bone (21/23) compared to kdm1a wild-type (18/18) larvae. Because the kdm1a mutant was a phenotypic, the originating lab chose not to pursue it further. To prevent others from spending redundant effort generating the same line, information about this mutant was submitted to ZebraShare (DeLaurier et al., 2020). KDM1A functions as a histone demethylase transcriptional corepressor in a multi-protein HDAC1/2/CoREST-containing complex (Hakimi et al., 2003; Shi et al., 2004). Humans with mutations in KDM1A are reported to have craniofacial defects including cleft palate and developmental delay (Tunovic et al., 2014; Chong et al., 2016); these clinical features are also found in Kabuki syndrome. In one study (Tunovic et al., 2014), clinical features are hypothesized to result from the combined effect of mutations in KDM1A and ANKRD11 (Ankrin Repeating Domain-Containing protein 11), the latter of which is associated with KBG syndrome involving craniofacial phenotypes. PHF21A and KDM1A interact, where binding of PHF21A to histones is required for the repressive activity of KDM1A (Lan et al., 2007; Kim et al., 2012). Given that both KDM1A and PHF21A underlie craniofacial defects in humans, zebrafish mutant models for these genes may be of potential interest to labs studying human syndromes such as Kabuki, KBG, and PSS-type syndromes.

Figure 4 Skeletal structure is normal in kdm1a maternal zygotic mutants.

Wild-type larvae (A–C) compared with kdm1a maternal zygotic mutant larvae (D–F). (A and D) Whole mount specimens, lateral view of head skeleton, (B and E) flat mount pharyngeal skeleton, pharyngeal arches 1 and 2, lateral view, (C and F) flat mount neurocranium, ventral view. Fish larvae are stained as described in Fig. 3. Scale bar = 200 microns.

Example 3, ctnnd1 mutants disintegrate by 24 hpf

The ctnnd1 gene was knocked out because it is within a locus associated with neuropsychiatric disorders (Schizophrenia Working Group of the Psychiatric Genomics Consortium, 2014) and the related gene CTNND2 has been linked to autism (Turner et al., 2015). Prior analysis using a ctnnd1 morpholino (MO) noted embryonic disassociation at high doses and bent tails at lower doses (Hsu et al., 2012); however, without mutant validation these ctnnd1-MO phenotypes are difficult to distinguish from non-specific morpholino toxicity (Stainier et al., 2017; Robu et al., 2007; Bedell, Westcot & Ekker, 2011). Consistent with a requirement for ctnnd1 in embryonic viability, in-cross of ctnnd1 heterozygotes yields no homozygous mutants 6 dpf (N = 44 wild type, 56 het, 0 mutant, Chi Square analysis P < 0.0001). Subsequent analysis revealed that the ctnnd1 homozygotes proceed through cleavage stages and gastrulate but typically die between the 4 and 12 somite stages (Figs. 5A and 5B). Embryonic death occurs via cellular disassociation (Fig. 5C), initiating at the head or the tailbud (Movie S1). In ctnnd1 heterozygotes, small clumps of cells briefly appear on the embryo’s dorsal surface during early somitogenesis stages, typically over the mid- or hindbrain regions (Figs. 5D–5F). Given that the homozygote fully disassociates, the heterozygote’s small clumps of ectopic cells may represent localized points of disassociation. Genotyping at 12 hpf confirms that 24/24 “dying” embryos are homozygous mutant, 25/25 embryos with ectopic cells are heterozygous, and 23/23 healthy embryos are wild type (Chi Square analysis P < 0.001). Because early lethality precludes analysis of neural phenotypes, information about the ctnnd1 mutants was deposited in ZebraShare (Thyme & Carte, 2020). Consistent with the zebrafish findings, the p120-catenin protein encoded by ctnnd1 has several known roles in early development, and murine loss-of-function models are embryonic lethal when homozygous (Hernández-Martínez, Ramkumar & Anderson, 2019). The mouse Ctnnd1 neural crest knockout line shows cleft palate when heterozygous, and consistent defects are seen in humans heterozygous for CTNND1 truncation alleles (Alharatani et al., 2020). Ctnnd1 is involved in cadherin stabilization, WNT signaling during gastrulation and epithelial-to-mesenchymal transitions, and suppression of the RhoA–ROCK–myosin pathway (Pieters et al., 2016; Yu et al., 2016; Hernández-Martínez, Ramkumar & Anderson, 2019). It is unclear which of these functions are the direct cause of the zebrafish ctnnd1 defects and researchers interested in any of these mechanisms and/or in CTNND1-related human disease may find this mutant useful.

Figure 5 Embryonic disintegration in ctnnd1 mutants.

(A, B) Frames from time-lapse imaging (Movie S1) illustrate disintegrating phenotype of ctnnd1 mutants. Asterisk(s) mark two of the embryos that disintegrated during the time-lapse recording. (C) DAPI staining of fixed ctnnd1 mutant embryos reveals that cells with intact nuclei dissociate from the embryo. (D–F) Phenotypes of ctnnd1 sibling embryos at the 6-somite stage. The homozygous mutant (F) has disintegrated, the heterozygous mutant (E) displays clumps of cells along the dorsal surface, and the wild-type embryo appears normal (D). The solid arrow marks a clump of cells dorsal to the midbrain and the open arrowhead marks a clump of cells dorsal to the hindbrain in the heterozygote.

Example 4, snu13a;snu13b double mutants arrest during somitogenesis

Previous analysis of Snu13 gene function in flies and zebrafish supported a specific role in muscle formation (Johnson et al., 2013; Williams et al., 2015). To further test this role, the two zebrafish genes, snu13a and snu13b, were knocked out. DNA sequencing of the resulting mutants indicates that they both cause frameshift and premature stop codons. The snu13a−∕− and snu13b−∕− single mutants are both overtly indistinguishable from wild-type siblings (not shown). Embryos homozygous for the two mutations (snu13a−∕−; snu13b−∕−) appear normal until 10 hpf but their development ceases to progress by 12 hpf, after which the cells linger in place and typically become necrotic by 24 hpf (Fig. 6). Some of the fish with this severe phenotype are homozygotically mutant for snu13b but only heterozygous for snu13a (snu13a+∕−; snu13b−∕−). These snu13a+∕−; snu13b−∕− fish are often indistinguishable from the double mutant shown in Fig. 6, but are sometimes indistinguishable from wild-type siblings (40 with wild-type phenotype were genotyped: 0 are snu13a−∕−; snu13b−∕−, 7 are snu13a−∕+; snu13b−∕−. 24 showing developmental halt were genotyped: 9 are snu13a−∕−; snu13b−∕−, 16 are snu13a+∕−; snu13b −∕−, Chi Square analysis P < 0.0001). These findings reveal that embryonic development can only sometimes proceed through somitogenesis stages when snu13 function is strongly reduced (snu13a+∕−; snu13b−∕−) and cannot proceed in the absence of snu13 gene function (snu13a−∕−; snu13b−∕−). A severe developmental halt is likely explained by a requirement for Snu13 protein in assembly of the core spliceosome (Stevens et al., 2001; Dobbyn & O’Keefe, 2004; Oruganti, Zhang & Li, 2005; Rothé et al., 2014; Diouf et al., 2018). These mutant phenotypes demonstrate that snu13a and snu13b gene function is essential to organismal viability and development past early embryogenesis. This severe embryonic phenotype impeded further investigation of muscle formation; however, these lines may be valuable to the broader research community, so information about the snu13a and snu13b mutants was submitted to ZFIN via ZebraShare (Gallati & Talbot, 2020). The mutants may be of interest to researchers investigating the core spliceosome, or as a comparison group for investigation of alternative splicing pathways. These mutants may also be useful to labs studying the maternal to zygotic transition because both snu13a and snu13b are expressed prior to zygotic genome activation (Papatheodorou et al., 2018).

Figure 6 Embryonic development arrests in snu13a;snu13b double mutants.

(A) Normal sibling and (B) snu13a;snu13b double mutant at 24 hpf. Tail region is outlined in orange, head in blue. The shown double mutant was confirmed to be homozygous by Sanger sequencing. Scale bar is 1 mm.

Discussion

Currently, there is no comprehensive zebrafish mutant library that parallels those available for yeast, drosophila, and worms (Winzeler et al., 1999; Thurmond et al., 2019; Harris et al., 2020). While large scale mutagenesis projects are being undertaken, such as the Sanger Targeting Induced Local Lesions in Genomes (TILLING) screen (Kettleborough et al., 2013), in most examples these mutant collections exist only in untested frozen sperm that carry many mutations in other genes. ZebraShare is intended to expand the ZFIN mutant collection by encouraging labs to publicize characterized alleles that lack an obvious route to traditional publication (Fig. 7A) and is a suitable destination for archiving information about alleles that labs may not plan to pursue.

Figure 7 A decision tree on whether to publicize findings in ZebraShare.

(A) A ZebraShare submission immediately disseminates your information to the scientific community and also can serve as a pathway to journal publication. (B) Comparison of correct and misplaced ZebraShare submissions, with solutions for incorrect uses.

Lines submitted to ZebraShare are publicly visible, and the abstracts can be cited, but the submissions may not include some information vital to journal publication, such as details about mutant construction. For instance, in this manuscript we cite abstracts of ZebraShare submissions but also include details about mutant construction in our methods section. To include these details, we needed to collect and coordinate information from each originating lab. Likewise, if other researchers would like to incorporate ZebraShare data into their own traditional publications, they should contact the lab which originated the mutant line. Thus, the ZebraShare system is intended to facilitate dissemination of unpublished mutant information and collaboration formation, to complement and enhance traditional routes of publication.

In this paper, we provide examples of mutants that we publicized using ZebraShare, such as a phenotypic alleles (kdm1a), mildly phenotypic (phf21aa), and those with severe early defects (snu13a;snu13b and ctnnd1). The diversity in described early lethal phenotypes, snu13a;snu13b arrests and ctnnd1 disintegrates, highlight the reality that different processes can underlie embryonic death. The observation that some snu13a+∕−; snu13b−∕− fish can proceed through somitogenesis while others cannot underscores variation often observed in mutant phenotypes, which is potentially influenced by genetic or environmental modifiers. New discoveries may stimulate new interest in submitted lines. While there was no clear disease connection when the ctnnd1 homozygous lethal phenotypes were submitted to ZebraShare, subsequent analysis of the heterozygote reveals a more specific cellular disassociation at the embryo’s dorsal edge, which could potentially be related to neural crest defects recently reported in humans with CTNND1 gene variants (Alharatani et al., 2020). Finally, ZebraShare may also provide information about alleles which reproduce phenotypes found already in publication and offers a way to quickly share phenotypes that verify or contradict the literature.

We generated ZebraShare to help researchers disseminate information about mutants which have no clear path to standard journal publication, including mutants that have no overt phenotypic defect. When mutants lack a desired phenotype, researchers may dismiss the finding because of compensatory mechanisms like gene redundancy, transcriptional compensation, unexpected splice variants, and maternal contributions (Ciruna et al., 2002; Rossi et al., 2015; Anderson et al., 2017; El-Brolosy et al., 2019). While these compensatory mechanisms sometimes do explain a lack of phenotype, the absence of phenotypic defect does not constitute evidence that one of these mutation-bypassing mechanisms are being used. In many cases a phenotypic mutants provide genuine insights into gene function. We strongly believe that the dissemination of information about such unexpected phenotypes is necessary to reduce duplicate effort and to foster honest, open discussion about the necessity, redundancy, and interactions between individual genes in zebrafish.

ZebraShare complements other rapid mutant dissemination platforms (Fig. 7B). For instance, CRISPRz allows researchers to share information about CRISPR guide RNAs but does not describe alleles generated nor mutant phenotypes (Varshney et al., 2016). Several researchers have put forward their own websites for describing mutants and transgenes (e.g., https://kawakami.lab.nig.ac.jp/), although individual lab websites may not be completely integrated into ZFIN. ZebraShare is conceptually similar to the ZFIN antibody and protocol wikis, which have provided valuable information to the zebrafish community for many years (Bradford et al., 2011; Howe et al., 2016). Unlike these wiki-style submissions to ZFIN, ZebraShare submissions are manually curated by ZFIN staff and are directly integrated into the database itself rather than as a separate wiki. ZFIN already accepts large datasets of less-characterized mutants and other direct submissions (Howe et al., 2016), which has been used by the Sanger TILLING project and several North American TILLING projects (Moens et al., 2008; Kettleborough et al., 2013 among others); however, ZebraShare is the first mechanism for labs to disseminate detailed information about individual mutations and phenotypes on ZFIN. Thus, ZebraShare fills a key niche by enabling individual labs to directly submit allelic and phenotypic information for up to a few mutants in ZFIN (Fig. 7B).

In the long term, ZebraShare will serve to facilitate reporting from our community’s collective project and enable the field to report about the function of more genes than can be communicated exclusively through traditional publications. We anticipate that researchers will contribute information about multiple alleles within individual genes as this information becomes available. Different lesions for single genes may have slightly different effects on RNA/protein (e.g., premature stop vs., splice error, vs. deletion of functional domains). Thus, deposition of information about multiple alleles will be extremely useful as our community discerns which mutation types have the strongest effects on development, and may influence the dialogue about discrepancies between morpholino and mutant data (Kok et al., 2015; Stainier, Kontarakis & Rossi, 2015; El-Brolosy et al., 2019; Tessadori et al., 2020). The ease of sharing will encourage examples and insights into how gene redundancy, maternal effect, and other ‘obscuring’ factors influence phenotypic severity. Furthermore, the simple ZebraShare submission process opens up opportunities for undergraduates, rotation students, and other new scientists to gain the transformative experience of describing and publicizing their findings in a formal and permanent manner with the broader community.

Conclusions

ZebraShare was conceived and developed in response to a community-wide need for a simple and centralized means to share information about alleles, particularly about a phenotypic lines (e.g., kdm1a), or mild or unexpected phenotypes (e.g., phf21aa). Yet, researchers may also want to submit mutants with strong and interesting phenotypes that are outside of the scope of their normal work. For instance, we show a role for ctnnd1 in embryonic integrity, and a role for snu13 genes in development past early somite stages. We anticipate that over time with community submissions growing, ZebraShare will be a valuable resource to facilitate active collaborations on submitted alleles, inform investigators of existing lines, provide preliminary information about potential roles of genes and variants of mutant alleles for those genes, and promote sharing and communication about mutant alleles within the field.

Supplemental Information

Supplemental Information 1 Original phf21aa sequences

Sequences for forward and reverse reads from WT and phf21aa mutant fish. These sequences are also found in GenBank, with accession numbers: wild type MW438986 and mutant MW438985.

Click here for additional data file.

Supplemental Information 2 Timelapse of ctnnd1 clutch during segmentation stages

Click here for additional data file.

We thank Mark Nilan and the University of Maine zebrafish facility for care of snu13aoz24 and snu13boz91 mutants, especially during the Coronavirus pandemic. We thank Sharon L. Amacher for supporting construction and early characterization of these snu13 mutants and the Ohio State Rightmire Hall zebrafish facility staff for initial care of these lines. We thank Alexander F. Schier for supporting construction and early characterization of the ctnnd1 mutants and the Harvard University zebrafish facility staff for initial care of these lines. JCT thanks Sara Loiselle (University of Maine) for comments on this manuscript and Jared Austin (University of Maine) for assistance with snu13a;snu13b genotyping. AD thanks Dr. Hyung-Goo Kim (Augusta University, Hamad Bin Khalifa University) for input and ideas about PHF21A. We also thank the ZFIN development team for creation of the ZebraShare feature in ZFIN.

Additional Information and Declarations

Competing Interests

Author Contributions

Animal Ethics

DNA Deposition

Data Availability

The authors declare there are no competing interests.

April DeLaurier and Jared C. Talbot conceived and designed the experiments, performed the experiments, analyzed the data, prepared figures and/or tables, authored or reviewed drafts of the paper, jointly had the original idea, early project development, and approved the final draft.

Douglas G. Howe and Leyla Ruzicka analyzed the data, authored or reviewed drafts of the paper, helped implement ZebraShare in ZFIN, and approved the final draft.

Adam N. Carte, Lacie Mishoe Hernandez, Kali J Wiggins, Mika M. Gallati, Kayce Vanpelt, Frances Loyo Rosado, Katlin G. Pugh, Chasey J. Shabdue and Khadijah Jihad performed the experiments, analyzed the data, authored or reviewed drafts of the paper, and approved the final draft.

Summer B. Thyme conceived and designed the experiments, performed the experiments, analyzed the data, prepared figures and/or tables, authored or reviewed drafts of the paper, and approved the final draft.

The following information was supplied relating to ethical approvals (i.e., approving body and any reference numbers):

IACUCs at The Ohio State University (2012A00000113), the University of Maine (approval: A2019_10_01), the University of South Carolina (approval number 2485-101478-031720), the University of South Carolina Aiken (approval: 010317-BIO-01), Harvard University (approval: 25-08), and the University of Alabama Birmingham (approval: 21744) approved this research.

The following information was supplied regarding the deposition of DNA sequences:

phf21aa sequences are available at GenBank: MW438986 (wild type) and MW438985 (mutant).

The following information was supplied regarding data availability:

Raw data are available in the Supplemental Files.

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
