# Peer review of "ZebraShare: a new venue for rapid dissemination of zebrafish mutant data"

_PeerJ, doi:10.7717/peerj.11007_

## Round 0.1 · original submission · Major Revisions

Thank you for submitting your manuscript to PeerJ. All three reviewers of your paper recognized the value of ZebraShare as a resource for the zebrafish research community. The novel data presented in the paper provide support for your explanation of why this resource is needed. Two of the reviewers have suggestions for improvement, with reviewer 1 providing extensive comments and expressing some concerns that the paper may not fit PeerJ’s publishing guidelines. I invite you to submit a revised manuscript after major revisions and ask that your rebuttal letter address all reviewer comments.

Reviewer 1 notes that your manuscript is clearly written and sees the value in ZebraShare as a new resource. They were unsure if the paper fits PeerJ’s editorial criteria (https://peerj.com/about/editorial-criteria/) as they did not feel that it included a hypothesis-driven research question. I believe that by explaining the gap in the research community’s ability to share information on generated mutants, and by presenting novel data on several mutants, you have met a portion of the editorial criteria. Reviewer 1 also has concerns about the details provided for each mutant and how methods are organized. PeerJ’s editorial criteria require that enough details be provided so that work is reproducible by another researcher, and reviewer 1’s concerns may stem from the lack of some of this detail. I note that in your paper you explain that some details are kept from ZebraShare submissions to encourage researchers that find mutants with the service to collaborate with the lab that originally generated them. If that is the reason for limiting details in your manuscript, you may want to address reviewer 1’s recommendation that you include fewer case studies but with more details. Please review our editorial criteria carefully and consider how your methods and results presentation can match the requirement for experimental details. Perhaps more details could be provided for a smaller number of case studies to support the argument for ZebraShare’s value. Please be sure to address these issues raised by reviewer 1, which I see as the primary ones for the paper.

I look forward to receiving your revised submission.

Reviewer 1 ·

Basic reporting

The authors describe ZebraShare, an online submission system as part of ZFIN that allows researchers to upload information on mutant lines that may be of general interest to the community. ZebraShare is currently live on ZFIN but no published details on how to submit information exists. The authors describe the workflow for submitting information to ZFIN and the various fields that are to be completed in the ZebraShare workbook prior to submission. The authors then provide five Case Studies of mutants that have been submitted to ZebraShare. The manuscript is clearly written and there is adequate background provided for the Resource.

Experimental design

The experimental design focuses on the workflow of submitting phenotype and mutant information into ZebraShare. The characterization of mutant data in their Case Studies is limited. The inclusion of several Case Study mutants is unclear and no obvious relationship exists between the different mutants. This particular manuscript focuses on a Resource (e.g. ZebraShare) rather than a specific hypothesis-driven research question. As such, there is some concern that this may not fully fall within the Aims of the Journal. While the analysis of the mutants following standard procedures the methods lacked adequate details (see below) related to the number of animals tested, whether techniques were performed on single embryos/larvae or pooled sample, and whether any statistical methods were conduced.

Validity of the findings

The benefits of ZebraShare to the zebrafish community include the dissemination of phenotypic information for mutants that may have limited benefit to the lab of origin. Perhaps the biggest benefit will that mutants lacking phenotypes can be identified to avoid duplicating negative results by multiple labs. It is debatable whether investigators would submit alleles that have novel phenotypes rather than attempt to follow-up. The workbook and submission process appears simple and straightforward. The online submission process is currently active and self-explanatory.

The inclusion of the Case Study mutants is less persuasive. These mutants have limited benefit to the labs of origin, but the inclusion of five different Case Studies is probably overkill. The data supporting the mutant characterization could be enhanced by additional information related to identify n-values, biological and technical replicates, and total numbers of animals analyzed.

Additional comments

The manuscript provides a description of ZebraShare and offers a brief description of submission process and the information required in the workbook. Both of these items are self-intuitive by looking at ZFIN and the workbook. Due to the simplicity of the submission process, the current manuscript describes a resource rather than documenting hypothesis-based science. One concern is that the submission process and workbook do not require detailed protocols and are rather self-explanatory. Unlike websites that require multiple toggles or explanation of the input s, this is a basic upload of an Excel spreadsheet. Dissemination of ZebraShare could occur over email announcements from ZFIN rather than publication here. The inclusion of multiple mutants in their Case Studies is also questionable. A single mutant would likely be sufficient. It’s not clear if the focus of this manuscript is on ZebraShare or the various Case Studies that lack the anticipated phenotypes. This detracts from the stated emphasis of the manuscript, which is explaining ZebraShare.

• The Methods section lists multiple sections on mutagenesis that are redundant and consists of both Methods and Results, with more emphasis on Results. It is not clear why the authors elect to say mutants were constructed as described in DeLaurier et al. and then proceed to give a lengthy description of mutagenesis results. The authors do not provide vendor information on enzymes, kits, of any concentrations for mRNA, gRNA, Cas9 protein. The text should be rewritten in an appropriate and concise format.
• The number of animals or larvae examined for various experimental techniques should be specified for all experiments. Statistical analysis, if necessary, should be performed.
• Within the methods, authors are encouraged to specify whether analyses, such as sequencing or PCR, were performed on single embryos or larvae or from pooled animals.
• The sentence in lines 247-249, “We do not intend to follow up with analysis of this defect, but this phenotype may be of interest to labs studying connective tissue in the head and face” is not relevant for this publication. The focus of the current manuscript is intended to be on the purpose of ZebraShare and the types of mutants that can be submitted.
• The choice of mutants included in this manuscript appear to have very little in common except their lack of utility to the authors. The authors are encouraged to provide some background supporting the inclusion of these particular mutants in this manuscript. While the use of ZebraShare is promoted as a means to provide information on unpublished mutants, a single mutant example would be sufficient in the current manuscript.
• Why did the authors elect to include five distinct Case Studies? In particular, Case Study 2 details the lack of craniofacial phenotypes in two unrelated mutants, the ldlrap1a and kdm1a mutants. It is not clear why the ldlrap1a mutant is included here at all. The human disease is hypercholerolemia. It is not mentioned whether this results in craniofacial phenotypes in humans? Why are the kdm1a mutants included with ldlrap1a in the Case Study here? There does not appear to be any functional or disease-associated relationship between these two genes. The authors would be encouraged to select a single mutant as an example of a ZebraShare submission.

Reviewer 2 ·

Basic reporting

No comment.

Experimental design

No comment.

Validity of the findings

No comment.

Additional comments

This study describes Zebrashare, a new resource hosted by Zfin that gives the opportunity for zebrafish investigators to share information about the mutant lines they have generated. Zebrashare was desperately needed and will without doubt be used extensively by the zebrafish community. Great study!

Reviewer 3 ·

Basic reporting

The current submission by DeLaurier et al. is a clear, concise, and well-written manuscript with very minimal typographical errors which I will enumerate below. They provided sufficient literature and background. The figures are all of high quality, and I find that the decision trees are especially informative in simulating the thought processes before, during, and after submission to ZebraShare.

Lines 104-105, 117-118, 132-133: Even though the cas9 mRNA information was described in the cited paper from the same group (DeLaurier, Alvarez & Wiggins, 2019), readers would really appreciate if this information was explicitly mentioned here.

Lines 144-145, 162-163, 175-176: Same as above. A little bit of detail here would be appreciated by the readers.

Delete Lines 409-411: These were repeated from Lines 406-408.

Figure 3 Title: For consistency, change "zebrashare" to "ZebraShare".

Figure 4 Delete "." before "Wild type"

Figure 5 Legend, after "microns". Delete "<!--[endif]-->"

Experimental design

Given the numerous targeted zebrafish mutants being generated around the world (including those generated via CRISPR-Cas techniques), there is a need to rapidly disseminate information on these mutants outside the confines of the traditional publication format. In this manuscript, the authors describe ZebraShare, a tool integrated within ZFIN, that fills this need. This new resource is relevant and a very useful tool for the whole zebrafish community. The authors had done an amazing job in defining the necessary information needed for each ZebraShare submission and in soliciting information on the validation of mutants. The mutants they described (case studies) in this manuscript are all appropriate, and they will help the readers and the zebrafish community in getting a sense of the many different kinds of mutants and the information that could be shared through ZebraShare.

Validity of the findings

The authors findings including the raw data provided are all robust and valid. Their conclusions are all well stated and are consistent with the case studies they presented.

Additional comments

Thank you for thinking about ZebraShare and for implementing it within ZFIN. It would be a very useful resource to the zebrafish community.

---

## Round 0.2 · accepted · Accept

Thank you for resubmitting your manuscript and your work to address the comments of the reviewers. All three original reviewers are satisfied that you have addressed their suggestions and I am happy to now accept your paper for publication in PeerJ.

Reviewer 2 has some minor suggestions that you could address in the final draft. I noticed that on line 251 the use of “provide submitted” seems redundant and should be corrected. I might also suggest changing “a-phenotypic” to “aphenotypic” throughout the manuscript.

You will be given the option to make the reviews of your manuscript available to readers. Please consider doing so as this review record can be a great resource for readers of your paper and contributes to more transparent science.

Thank you for choosing PeerJ as a venue for publishing your work. I am excited that our journal will announce this valuable new resource for the zebrafish research community.

Reviewer 1 ·

Basic reporting

No comment

Experimental design

No comment

Validity of the findings

No comment

Additional comments

The authors have made substantial revisions that have significantly improved the manuscript and address the prior concerns. ZebraShare should be a useful resource for the community.

Reviewer 2 ·

Basic reporting

This revised study describes Zebrashare, a new resource hosted by Zfin that gives the opportunity for zebrafish investigators to share information about the mutant lines they have generated. Zebrashare was desperately needed and will without doubt be used extensively by the zebrafish community.

Experimental design

The authors have added details in their description of their approaches and have included quantifications and statistics to the results. The focus on 4 different examples (instead of 7 in the original version) describing mutants with variable phenotype improves the manuscript and makes it easier to read.

Validity of the findings

The data presented by the authors are clear and convincing.

Additional comments

This revised manuscript by DeLaurier et al., is well written and describes a new resource hosted by Zfin that was desperately needed by zebrafish investigators. It will give them the opportunity to share information about the mutant lines they have generated and possibly develop new collaborations.

This new version addresses the initial comments made by reviewers. I only spotted a few typos and minor details that need correction:
- line 143: please indicate the origin of Cas9 protein
- line 159: add the reference Parant el al., 2009 (Dev. Dyn) for describing HRMA
- line 116: write mMESSAGE instead of mMessenge
- line 155: replace with “we synthesized the Cas9 mRNA from ….”
- lines 284 and 319: add commas after last names in references
- line 347: add an “s” to “highlight”

Reviewer 3 ·

Basic reporting

The authors have addressed all my comments.

Experimental design

No comment.

Validity of the findings

No comment.